# A Multi-Considered Seed Coat Pattern Classification of *Allium* L. Using Unsupervised Machine Learning

**DOI:** 10.3390/plants11223097

**Published:** 2022-11-14

**Authors:** Gantulga Ariunzaya, Shukherdorj Baasanmunkh, Hyeok Jae Choi, Jonathan C. L. Kavalan, Sungwook Chung

**Affiliations:** 1Department of Computer Engineering, Changwon National University, Changwon 51140, Republic of Korea; 2Department of Biology and Chemistry, Changwon National University, Changwon 51140, Republic of Korea; 3Department of Computer and Information Science and Engineering, University of Florida, Gainesville, FL 32611, USA

**Keywords:** *Allium* seed coat, testa sculpture, unsupervised machine learning, SEM, new grouping

## Abstract

The seed coat sculpture is one of the most important taxonomic distinguishing features. The objective of this study is to classify coat patterns of *Allium* L. seeds into new groups using scanning electron microscopy unsupervised machine learning. Selected images of seed coat patterns from more than 100 *Allium* species described in literature and data from our samples were classified into seven types of anticlinal (irregular curved, irregular curved to nearly straight, straight, S, U, U to Ω, and Ω) and five types of periclinal walls (granule, small verrucae, large verrucae, marginal verrucae, and verrucate verrucae). We used five unsupervised machine learning approaches: K-means, K-means++, Minibatch K-means, Spectral, and Birch. The elbow and silhouette approaches were then used to determine the number of clusters required. Thereafter, we compared human- and machine-based results and proposed a new clustering. We then separated the data into six target clusters: SI, SS, SM, NS, PS, and PD. The proposed strongly identical grouping is distinct from the other groups in that the results are exactly the same, but PD is unrelated to the others. Thus, unsupervised machine learning has been shown to support the development of new groups in the *Allium* seed coat pattern.

## 1. Introduction

The genus *Allium* L. is the largest genera in the family Amaryllidaceae [1,2,3] and consists of more than 1000 species [4] distributed in the northern hemisphere and southern Africa [1,4]. The most recent *Allium* taxonomy consists of 800 species, 15 subgenera, and 56 subsections [1]. Several research groups worldwide are studying *Allium* species. Depending on their macro- and micro-morphological, molecular information, biogeographical distribution, and evolutionary history, some ambiguous subgenus and sections have been updated [1]. Therefore, scientists and researchers have begun to investigate the morphological, anatomical, developmental, and phytochemical characteristics of a variety of *Allium* taxa.

Seed coat pattern is one of the most important taxonomic features that support taxonomic relationships in *Allium* [5,6,7,8,9,10]. The species level typically exhibits a high degree of diversity, and infraspecific diversity is also significant [8,9,10,11]. According to Yusupov et al. [3], the macro- and micro-morphology of seeds are one of the most important taxonomic features that delimit the taxa in *Allium*. Moreover, Celep et al. [9] identified testa cell form, shape, and sculpturing of periclinal walls, and the position, shape, and type of undulation of anticlinal walls as essential diagnostic features used to categorize taxa at the sectional level. Seed testa sculpture attributes in combination with seed shape provide key features for distinguishing major clades of *Allium* in molecular phylogeny [1]. In general, there are two types of walls: anticlinal wall undulation includes irregularly curved, irregularly curved to nearly straight, Ω, U, U to Ω, S, and straight, whereas periclinal wall undulation includes granule, small verrucae, large verrucae, central verrucae, marginal verrucae, and dense granule [5,6,7,8,9,10,11,12,13,14,15,16]. 

Recently, identification and classification based on deep learning and machine learning have greatly influenced several research disciplines. One of them is the field of plants. Some studies use plant disease and flower classifications already trained using convolutional neural networks (CNNs) and transfer learning to conduct new studies and improve the productivity of previous works. For example, Geetharamani [17] used a trained CNN to identify an open dataset of plant leaf diseases with 39 different classes. He used CNN to extract valuable leaf features directly from the raw representations of the input data and used a deconvolutional network technique to gain an understanding of the selected characteristics [18]. Reyes et al. [19] pre-trained a CNN using 1.8 million images and applied a fine-tuning technique to transfer the learning recognition capabilities from general domains to the unique issue of plant identification. Researchers prefer machine-learning-based techniques over human-based classification because deep learning and machine learning have several advantages with specific data.

In recent decades, machine learning and deep learning have been successfully applied in biology for classification, segmentation, and identification. For example, Saleem et al. [20] proposed two steps to classify plant diseases using a CNN. First, they compared well-known CNN architectures with modified and cascaded/hybrid versions of some of the deep-learning models proposed in recent works to determine the CNN that performed best. They then evaluated how well the best model performed after training with different deep-learning optimizers. In addition, Wang et al. [21] used real-time detection of tomato diseases in the environment based on YOLOv3-tiny network architecture. These research methodologies are distinguished by their use of labeled datasets. These datasets are used to train or monitor algorithms to accurately identify data or predict outcomes. With labeled input and outputs, the model can measure its accuracy and learn over time. In contrast, we looked at unsupervised machine learning, where unlabeled datasets are analyzed and clustered. The main idea is to discover hidden patterns in the data without requiring human intervention. 

In recent years, researchers have become interested in plant identification and classification through machine and deep learning. Several studies used machine-learning and deep-learning classifiers to classify plant diseases by processing leaf and seed images. Most researchers used support vector machines to classify grape plant diseases [22,23], as well as neural networks [24]. In addition, the deep-learning model of the Efficient model can classify plant leaf disease [25,26]. Deep learning was used to classify tomato leaf diseases [27], and banana leaf diseases [28]. However, these studies are only based on the classification of plant diseases using machine learning such as deep learning. 

The identification and classification of various seed coat patterns using machine learning and deep learning have not been fully studied. Similar to our study, a new class of coralline algae was created by fine-tuning pre-trained CNNs [29]. They classified four commonly occurring Mediterranean species (*Lithothamnion corallioides*, *Mesophyllum philippii*, *Lithophyllum racemus*, and *Lithophyllum pseudoracemus*) at the species and genus levels. Therefore, we describe how to determine our own grouping based on the seed coat pattern using machine learning.

In this study, we used unsupervised machine-learning techniques to classify a set of scanning electron microscopy (SEM) images of *Allium* seed coat patterns. We used the following techniques: K-means, K-means++, Minibatch K-means, Spectral, and Balanced Iterative Reducing and Clustering utilizing Hierarchies (Birch). Because of their ability to discover similarities and differences in data, they are an excellent solution for exploratory data analysis. We present how to form groups using unsupervised machine learning. 

We proposed to classify seed coat patterns using unsupervised machine learning.We then compared them to previous human-based classifications.Following that, we suggested our proposed classification based on unsupervised machine learning and possible combinations such as SI, SS, SM, NS, PS, and PD.

## 2. Results 

In this section, we report two types of results comparisons: a comparison between the results of human- and machine-based classification methods, and a comparison between the results of our combinations. This allows us to determine the relationship between each method. We selected three types of threshold values for data scales. First, we selected a minimum threshold value of 500 because we believed that it could provide useful information and was a reasonable value. We then selected a medium value between 1000 and 2000, and a maximum value above 2000. By increasing the threshold, we strongly expected that the results would be the same.

### 2.1. Comparison between Results of Human-Based and Machine-Based Methods 

We first report the results for each relationship in tables provided for the machine-based cluster. As briefly explained in Figure 1, we named the relationship of each group. That is, (1) method name, (2) cluster name, and (3) separation number. The method and cluster are present in this case, but since one method depends on another, it must be separated again. If the threads M1C1-3 and M3C1 match M3C1 and M4C3 in Figure 2a, it means that M1C1-3 is connected to M1C1 three times. In addition, M1C1 and M2C1-2 match with M2C1-2 and M4C3 in Figure 2b. Figure 2d shows the correspondence between M1C2 and M3C1 and M2C2 and M3C3. Therefore, this relationship can be considered as a cluster image of comparable threads. 

Figure 2b shows that the numerical value for identical M2C3-1 and M3C1 and M2C3-2 and M3C2 ranges from 1120 to 1127. Moreover, the numerical value for identical models, M2C1 and M2C1-1, M2C2 and M3C1-2, M2C3 and M2C1-1, M2C4 and M3C2-2, M2C5 and M3C3, M2C5 and M3C6, M4C1 and M5C4-1, M4C5 and M5C1-2, M4C6 and M5C1-2′s in Figure 2c ranges from 1120 to 1127. Therefore, these images can be considered identical. However, there is no relationship in Figure 2c,d. In Figure 2d, only a relation of 1312 is shown.

Related M2C3 and M3C3 in one thread that corresponds to M3C1 and M2C3 in Figure 3c. In both groups Figure 3a,b, there is no connection and no thread.

For the thread, five-colored clusters of ST, associated with M1C2 and M3C1 correspond to M2C2 and M3C3 in Figure 4d. Therefore, this model provides reasonable support for clustering and very similar clusters. 

For identical: M2C2 and M3C1-1, M2C2-3 and M3C2-1, M2C3-1 and M4C6-1, M2C5-1 and M4C4, M2C5-2 and M4C5, M2C2 and M3C1-2, M2C1 and M5C5, M2C2 and M3C1-2, M2C2-4 and M5C2, and M2C3-2 and M4C2 in Figure 4b. Moreover, the values of M2C2-2 and M3C1-1, M2C2-3 and M3C2-1, M2C3-2 and M4C6-2, M2C5-1 and M4C4, M2C5-2 and M4C5, M2C2-1 and M5C5, M2C2-4 and M5C2, and M3C2-2 and M4C2 are between 911 and 912, which means that they are very similar images. For Figure 4a,c, there is no relation and similar values, only 4d shows at least one relation.

### 2.2. Proposed Clustering

In this section, we present the main clustering generated by a machine. For this purpose, we used grayscale, threshold, and colored image clustering through the elbow and silhouette methods. Using these methods, we compare the number of cluster images that can represent a relation and a thread between these types of techniques and determine the number of images that can be clustered together. Table 1 presents the abbreviations and shows how we define them in this paper. Table 2 presents the main clustering of machine-based observations. In addition, the data in Table 2 belongs to Figure 5 and Figure 6.

Figure 5 illustrates the model of the proposed clustering. As for the threads, related EG-C1 and EC-C1 correspond to EG-C4 and SC-C1 correspond to EC-C1 and SC-C1. This approach seems to suggest that these clusters are similar threads. Figure 6 shows detailed information about each group. Therefore, this relation is more than one of anticlinal and periclinal wall threads.

For the identical approach, the numerical value of EG-C4 and SC-C1 and EC-C1 and SC-C1 is 845 and that of EG-C1 and EC-C1 and SG-C1 and ET-C1 is 1127. These values are equal to the anticlinal straight result of the six-grayscale clusters and four-threshold clusters. 

Therefore, we have mentioned the most important observation regarding machine-based clustering. In this case, where there is more than one thread of human-based clustering, we can see that the thread, EG, SC-C1, and EC-C1′2 images may be the same. In these cases, the information given by the two thread approaches was similar due to the similarity of the information from the threads.

Next, we determine the type of image used. For Figure 7, the five most prevalent images were selected from each image in the six-method testing. This allows us to determine the group the images came from. The result is that the images from 7a to 7d are similar, the images from 7e are slightly distinct, and the images from 7f are largely different from the others.

## 3. Discussion

Many researchers have studied the deep- and machine-learning-based classification of plant organs such as plant disease classification [25,26], and tomato-type classification [27]. However, there is no research on the identification and classification of seed coat patterns using machine learning or deep learning to date. The seed testa sculpture is one of the most important taxonomic features for plants. Many researchers determined seed coat shape patterns, sculpturing of the periclinal wall, and type of undulation of anticlinal walls to identify species of *Allium* [1,2,5,6,7,8,9,10,11,12,13,14,15,16]. However, humans identify different characteristics of seed coat sculptures. Therefore, we identified the seed coat pattern of *Allium* images using deep learning for the first time. 

In our approach, the use of unsupervised machine learning methods has provided the opportunity to investigate new types of morphology, including ultrastructure. We attempted to determine our own clustering, which provides another proposal using unsupervised machine-learning techniques for the clustering of *Allium* seeds. We presented the proposed analysis by showing the quantities of each group in Figure 5 and the images that best fit our criteria in Figure 6 and Figure 7.

We made comparisons between all models and our model. In the highlighted analysis, we show the kind of images included in each case. We obtained five common images from each group. In the SI case, 540 images were from the EC-C1 and SC-C6 group, whereas 845 images were from the EC-C1 and SC-C1. The SS case contained 869 images from the SG-C1 and ET-C1 groups. As for the SM case, it contained 1127 images from SG-C1 and ET-C1. The EG-C1 and EC-C1 images were taken from 1127 images in the NS group. In addition, the EG-C1 and SC-C1 groups contributed 1127 images. Another 845 images were taken by the EG-C4 and SC-C1 in the PS group. The PD group also included 824 images from EG-C4 and SC-C4. 

The second part of our analysis compared our final group with the five methods used. Because the five-method group was obtained using human-based clustering it showed overlapping anticlinal for straight, U, and U to Ω, and periclinal wall types for marginal verrucae and small verrucae. Indeed, the two relations shown in Figure 6 are proposed approaches. In contrast, human-based clustering showed one relation in Figure 2a,b, Figure 3c and Figure 4d. The reason we mention the previous association is that we want to highlight our own clustering for a distinct concept other than conventional classification. Using the two prior cases, we then also identified a similar case. The value of small verrucae in the four-threshold group’s count value for the human-based group value ranging from 1120 to 1127. In addition, the count value of the marginal verrucae six-grayscale case was between 911 and 912. The count value for EG-C4 and SC-C1 and EG-C1 and SC-C1 was between 1120 and 1127 and that for EG-C1 and EC-C1 and SG-C1 and ET-C1 was 845. The identical count value for the final group was the same as for the first five method groups. We discovered SI, SS, SM, NS, PS, and PD using the proposed approach. The SI group outperformed the SS, SM, NS, PS, and PD groups. In addition, the SS, SM, NS, and PS groups are very similar. Moreover, the PD groups are still distinct. 

## 4. Materials and Methods

### 4.1. The Proposed Module Overviews

The proposed algorithm was implemented by grouping the data in an unlabeled dataset based on the underlying hidden features in the data in Figure 8, which consisted of two phases: (a) general module and (b) performance analysis module. These two phases are explained in more detail in the following subsections.

### 4.2. General Module

#### 4.2.1. Datasets

We selected the seed coats of over 100 *Allium* species from previous studies [5,8,9,10,30]. The seed coat sculptures of *Allium* species are displayed in Figure 9. The species names along with descriptions of anticlinal and periclinal walls as well as corresponding references of all selected species are given in Appendix A. 

Figure 9 depicts seed coat types for anticlinal and periclinal walls. Anticlinal walls are the boundaries between two perpendicular cell walls, whereas periclinal walls are the cells outside the surface [31]. Among the seeds depicted in the picture, we selected three and two characters from the anticlinal and periclinal walls, respectively. The most overlapping types in the test results were straight, U, U to Ω from the anticlinal wall and marginal verruca, small verruca from the periclinal wall. 

#### 4.2.2. Preprocessing Module

To prepare the image for the main system operation (the image in Figure 8), preprocessing is required to deal with varied input data sizes, noise, and color. First, the goal is to crop the seed walls separately (Figure 10a–f). This ensures dimensional uniformity in all images used in the next step. Each image is normalized at this point, by cropping it to 250 × 250 pixels, to subsample it into the desired dimension. Preprocessing involves images such as grayscale images (Figure 10i), threshold images (Figure 10k), and colored images (Figure 10l). For the first preprocessing of the grayscale image, the input image was converted to a grayscale image. Grayscale representations are often used to extract descriptors rather than color images directly because grayscales reduce computational costs and simplify algorithms [32,33]. Color might be of limited interest in many applications, and the introduction of unnecessary data might increase the required training data. In the following preprocessing stage, the pixel values were divided into white backgrounds or black foregrounds using a threshold [34]. 

Thresholding is the simplest image segmentation method and the most common way to convert a grayscale image to a binary image. In the next step, we selected a threshold value, and all the gray level values smaller than the threshold value were classified as 0 (black, or background) while all the gray level values equal to or greater than the threshold value were classified as 1 (white, or foreground). It is important to remember that grayscale images contain pixels with values between 0 and 1, therefore the threshold value is within a closed range [0.0, 1.0] [35,36]. In this case, we set the threshold to 0.6. The pixel larger than 0.6 indicates an original pattern in white, and the remaining pixels indicate a background pattern. In the final preprocessing of the colored image, we applied the label-connected component to the threshold image. The label-connected component associated with the label has the constraint of accepting a binary image [37,38]. It can also support a label matrix which is usually output from a label component connected to the label as well as binary images. This binary image should contain a set of objects that are separated from each other. Pixels that belong to objects are labeled 1 and pixels that are in the background are labeled 0. In fact, an object is those pixels that are 1 that are connected in a chain when considering local neighborhoods [37]. It gives the affiliation of each pixel. This indicates where each pixel belongs if it falls on an object. In the final preprocessing phase, the applied measurement measures the different image quantities and features in a black-and-white image. More specifically, based on black and white, it automatically determines the properties of each contiguous white region that is eight-connected. The purpose of the final stage was to visualize the color assigned to each object based on the number of objects in the label matrix. Once the preprocessing phase is complete, the images are immediately passed on to the next process.

#### 4.2.3. Augmentation Module

To increase the diversity of the dataset, a suitable data augmentation technique is needed to improve the size and quality of the training set by creating modified data from the existing data [39,40]. The purpose of this technique is to extend and improve the dataset to reflect deformations in diverse seed patterns in a real-world setting. In our study, random rotation was performed with a maximum angle of 5°, width and height shifts were performed with a value of 0.01, and shifting pixels from left to right and top to bottom was performed.

#### 4.2.4. Shuffling Raw Data

We then split the data into separate training and testing sets. Before training the machine learning model, it is important to thoroughly shuffle the dataset to avoid bias or patterns within the split dataset [41]. The goal of dataset shuffling is to improve the quality and predictive performance of machine-learning models. In this study, we randomly shuffled the training dataset.

#### 4.2.5. Calculating the Number of Clusters

One of the biggest challenges in unsupervised learning is determining the number of clusters needed. Therefore, it is necessary to determine the number of clusters in advance of clustering [42,43]. We applied the elbow and silhouette schemes, which are useful techniques in evaluating the quality of clustering, to determine the optimal number of clusters in our study. The elbow method is easy to implement because the k number of clusters is based on the sum of squared distance (SSE) between the data points and their associated cluster centroids [44,45,46]. In the silhouette technique, unlike the elbow method, the silhouette coefficient is calculated and the number of clusters is easily determined [46]. The value of the silhouette coefficient ranges from [−1, 1], where a high score indicates that the sample is far from the neighboring clusters and a low score indicates that these samples may have been assigned to the wrong cluster [47]. Thus, we applied the elbow and silhouette schemes to determine the optimal number of clusters, changing the value between 2 and 10. Figure 11a shows the original cluster created by human-based clustering. In contrast, Figure 11b shows the number of clusters obtained by the elbow method, and Figure 11c shows the number of clusters obtained by the silhouette method. 

#### 4.2.6. Clustering Module

The goal of unsupervised methods is to determine whether a group of data is formed based on similarities between individual pieces of information. Consequently, cluster analysis is an excellent method to study the relationships between groups. In this study, we used K-means, K-means++, Minibatch K-means, Spectral, and Birch clustering, which are extremely easy and among the most popular unsupervised machine-learning methods. It may be difficult and arbitrary for a machine to decide the number of clusters to form in K-means, one of the most commonly used algorithms for cluster analysis [48], when the number of clusters is given in advance. For clustering analysis, we used K-means clustering to compute cluster indices, centroid locations, and distances between points and centroid positions. Next, we introduced a popular variant of the classic K-means algorithm, named K-means++. Bahmani et al. [49] demonstrated how this algorithm can significantly improve the quality of clustering by improving the initialization of centroids. Therefore, we also used K-means++ and included widely separated centroids. This increases the likelihood of initially picking up centroids that are in different clusters, and since the centroids are picked up from the data points, each centroid ends up with some data points associated with it. There is a version of the K-means algorithm in unsupervised machine learning known as Minibatch K-means. The algorithm creates random batches of data for storage in memory and then collects a random batch of data at each iteration to update the cluster [50]. It also facilitates cluster search by reducing computational costs. It is also a widely used technique in spectral cluster analysis. Spectral clustering uses a connectivity approach for clustering where communities of connected or immediately adjacent nodes are identified graphically [51]. The nodes are then mapped into a low-dimensional space that can be easily separated to form clusters. The Birch clustering algorithm can cluster a large dataset by first creating a small and compact summary of the large dataset that contains as much information as possible [52]. It uses hierarchical methods to cluster and reduce data. Birch only needs to scan the data set in a single pass to perform clustering [52]. 

### 4.3. Performance Analysis

In this section, we show a bi-directional representation of the community using our proposed performance analysis. We used five types of unsupervised clustering analyses to characterize overlapping relationships. We first introduce how to evaluate the performance analysis results of data groups or clusters, which consist of two types of analyses: the comparison between human- and machine-based analysis, and our main clustering. Table 3 shows the abbreviations used in our study.

We first illustrate the combinations algorithm for overlapping relation needed in this study in Figure 12.

Figure 12a shows a comparison between the methods used to analyze relationships. In this section, we explain how we use this algorithm. First, we read the test image of the raw data for performance analysis. Then, we compare the results of the methods we find related. For example, it would be useful to compare M1C1 and M1C2, M1C1 and M1C3, and M1C1 and M1C4 as shown in Figure 12a. This process will be completed after M5C4. Thereafter, the number of times overlapping values occurred is recorded. Figure 12b illustrates the main analysis. It is the same process as before. However, the comparison procedure is different from that represented in Figure 12a. The difference is that we used the elbow and silhouette method to determine the number of clusters required. We found several types of clusters. Therefore, we compared EG and SC, EG and ET, EG and ST, EG and EC, EG and SC. 

#### Visualization Module

After completing the performance analysis, we introduce the visualization module, which we used to connect the entire result of cube-based visualization. In the cube-based visualization, the number of cubes we need is obtained from Equation (1). Equation (2), then gives the amount of the tables. In the formula, *k* represents the number of possibilities for the selected objects, ! represents factorial, and *n* represents the total number of samples in the set.

Mathematically, the formula for determining the number of arrangements by selecting only a few objects from a set without repetition can be expressed as follows: (1)$$C_{n}^{k}=\frac{n!}{k!\left(n-k\right)!}$$

Therefore, the total number of tables based on the preceding methodology is as follows: (2)∑tables

## 5. Conclusions

This study used unsupervised machine learning to categorize the seed coat patterns of *Allium* species. We then compared the categories with previous human-based classifications. Following that, we proposed and examined unsupervised machine learning and possible combinations, which are SS, SI, SM, NS, PS, and PD. The anticlinal wall type of straight obtained the best results, with four-grayscale clusters and four-threshold clusters, while the anticlinal wall type of U to Ω produces good results with four-grayscale clusters. Furthermore, the periclinal wall of small verrucae five-colored cluster produces the best result. There was one anticlinal and periclinal wall relationship that was strongly related. The proposed SI group outperformed the SS, SM, NS, PS, and PD groups, whereas the SS, SM, and NS outscored the PD group. Therefore, unsupervised machine-learning algorithms were discovered to be suitable for grouping seed coat patterns.

## Figures and Tables

**Figure 1 plants-11-03097-f001:**
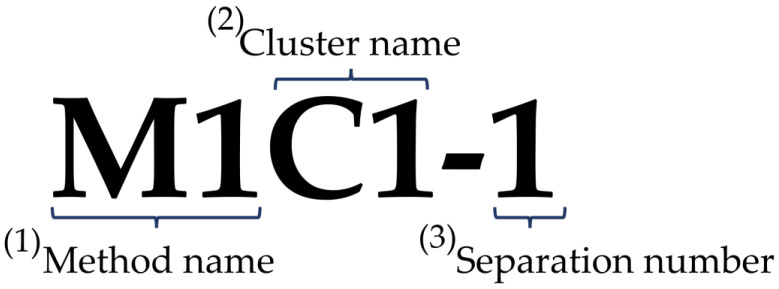
Example relationship of the group for separations.

**Figure 2 plants-11-03097-f002:**
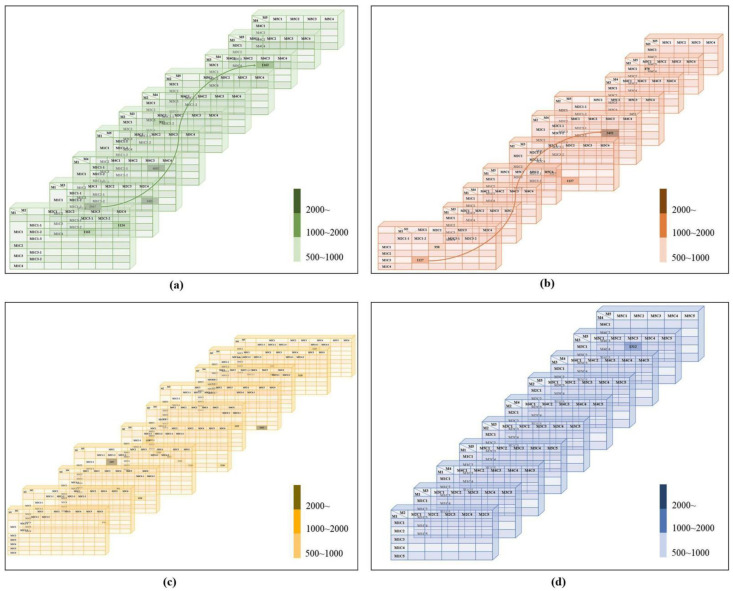
Anticlinal wall result–straight results. (**a**) four-grayscale clusters result, (**b**) four-threshold clusters result, (**c**) six-grayscale clusters result, (**d**) five-colored clusters result.

**Figure 3 plants-11-03097-f003:**
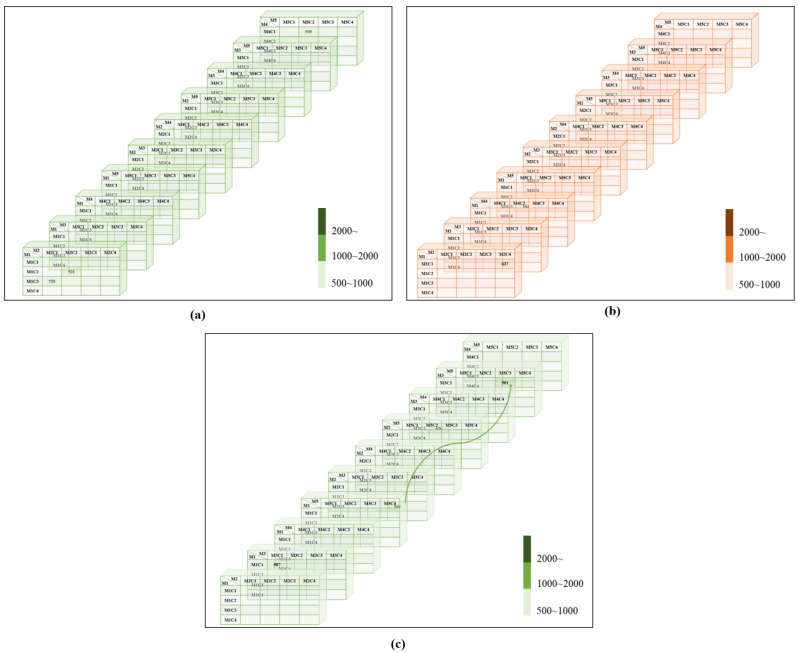
Anticlinal wall results. U to Ω—(**a**) four-grayscale clusters result, (**b**) four-threshold clusters result, U—(**c**) four-grayscale clusters result.

**Figure 4 plants-11-03097-f004:**
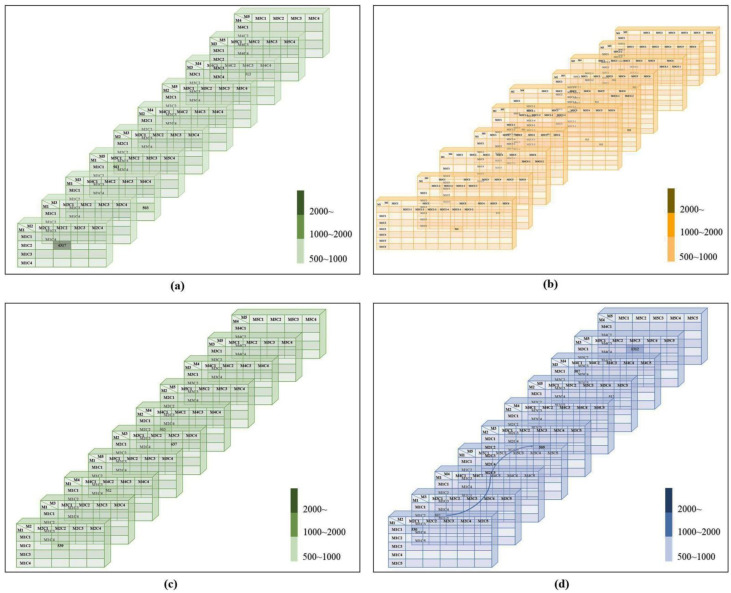
Periclinal wall results. Marginal verrucae—(**a**) four-grayscale clusters result, (**b**) six-grayscale clusters result, Small verrucae—(**c**) four-grayscale clusters result, (**d**) five-colored clusters result.

**Figure 5 plants-11-03097-f005:**
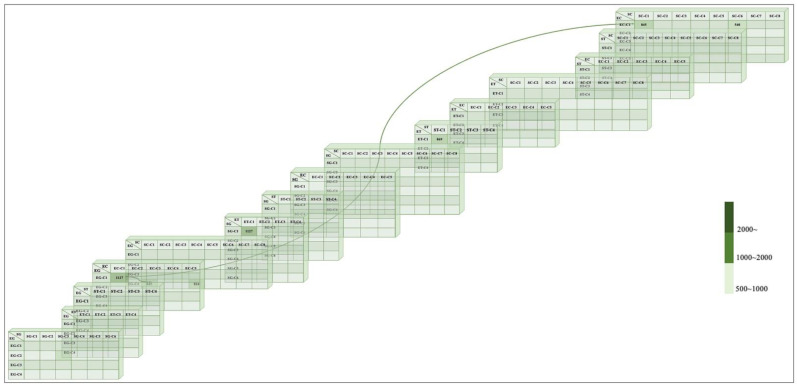
Proposed clustering.

**Figure 6 plants-11-03097-f006:**
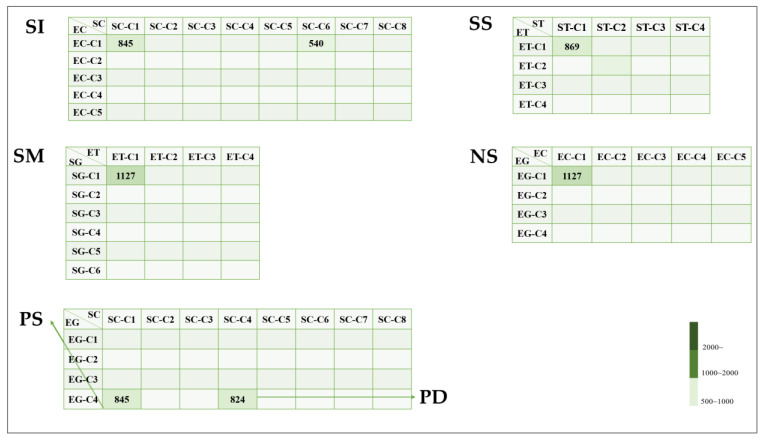
Detailed information on the proposed clustering.

**Figure 7 plants-11-03097-f007:**
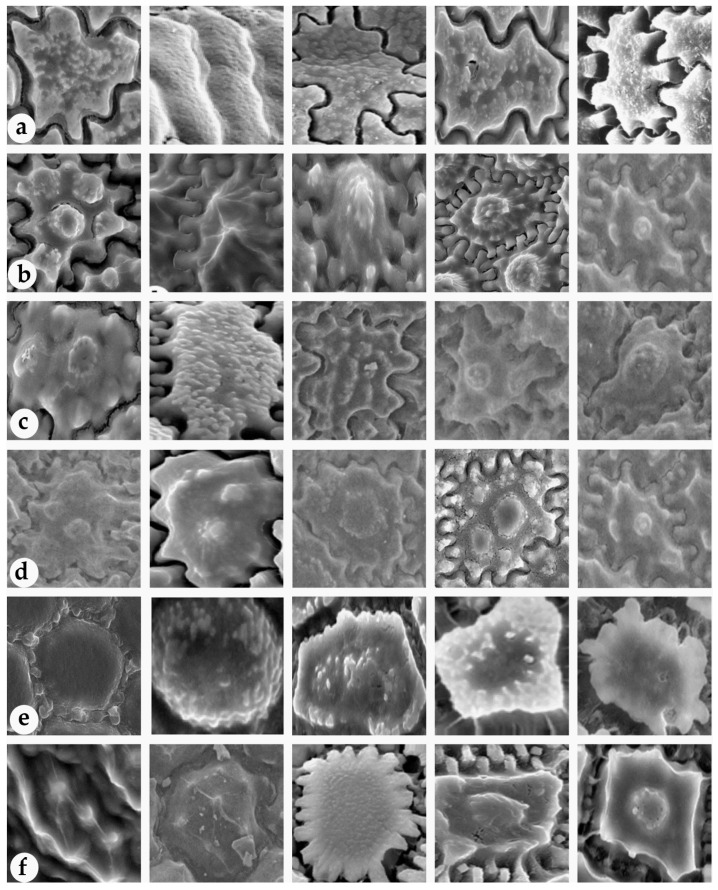
Related images for each group from [5]. (**a**) Actual images in SI, (**b**) Identical image in SS, (**c**) Similar image in SM, (**d**) Approximately similar image in NS, (**e**) Possibly similar image in PS, (**f**) Remaining images in PD.

**Figure 8 plants-11-03097-f008:**
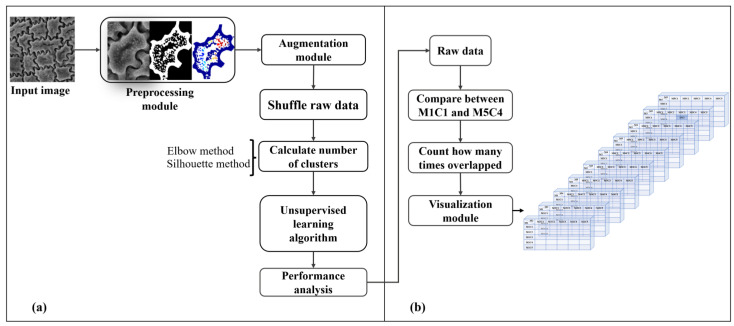
Proposed algorithm for this study. (**a**) General module, (**b**) Performance analysis module.

**Figure 9 plants-11-03097-f009:**
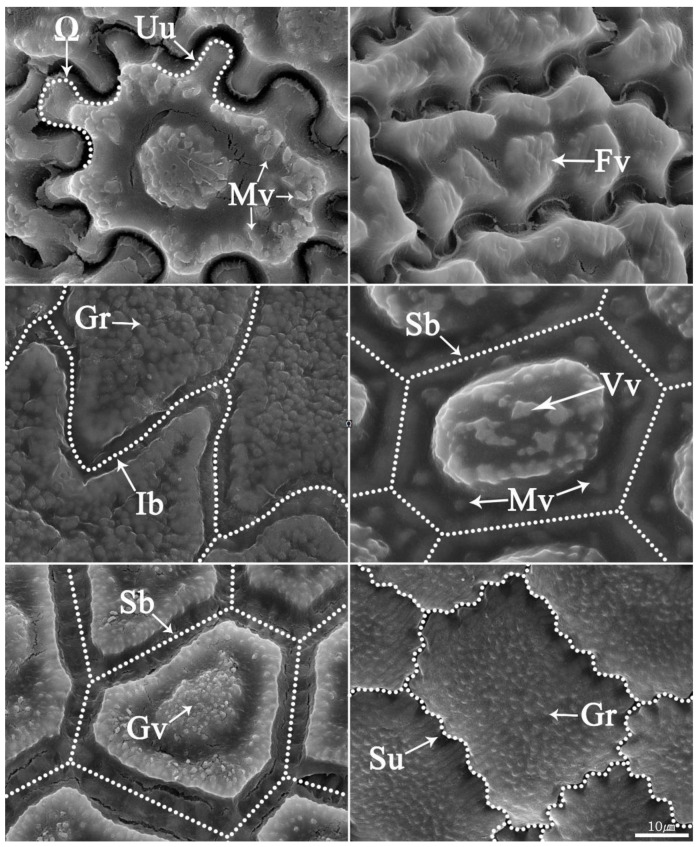
Type of seed testa of *Allium* derived from [5]. Ω, Omega-type undulation; Uu, U-type undulation; Su, S-type undulation; Ib, irregular boundary; Sb, straight boundary; Gr, granule; Vv, verrucate verruca; Mv, marginal verruca; Gv, granulate verruca.

**Figure 10 plants-11-03097-f010:**
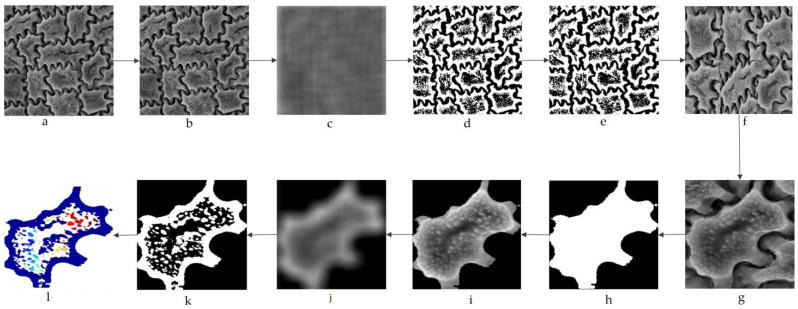
Data preprocessing of the seed coat pattern. (**a**) Input image, (**b**) grayscale image, (**c**) threshold using adaptive thresholding, (**d**) binary image, (**e**) labeled image (**f**) and cropped image, (**g**) separated cropped image, (**h**) masked image, (**i**) grayscale image, (**j**) binary image obtained using an adaptive threshold, (**k**) binary image, (**l**) and colored image.

**Figure 11 plants-11-03097-f011:**
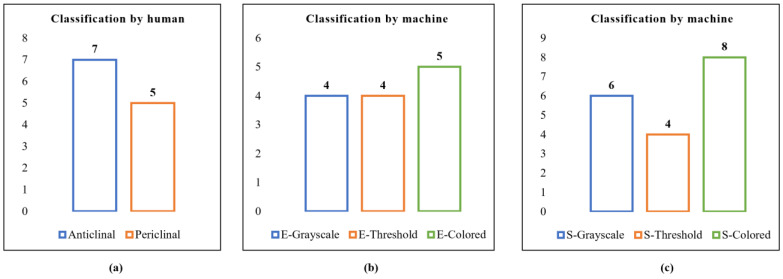
(**a**) Original number of clusters generated by humans. (**b**) Number of clusters generated using the Elbow method. (**c**) Number of clusters generated using the Silhouette method.

**Figure 12 plants-11-03097-f012:**
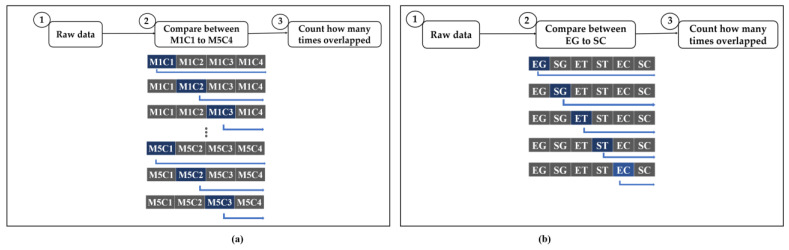
Overlapping algorithm overview. Each method generates overlap relations as well as threads based on the results. (**a**) Comparison between human-based and machine-based algorithm (**b**) Our proposed algorithm.

**Table 1 plants-11-03097-t001:** Proposed picking grouping.

Our Observations	Abbreviation	Definition
Strongly identical	SI	It refers to an image that is strongly similar.
Strongly similar	SS	It is referring term refers to an image that is exactly similar.
Similar	SM	It refers to one or more images that are similar to one another.
Nearly similar	NS	It indicates that two or more images are similar to one another.
Possibly similar	PS	It indicates that there is a chance that certain images are identical.
Possibly different	PD	It is indicating that there is a chance that some images may be different.

**Table 2 plants-11-03097-t002:** Related image number for each group.

SI	SS	SM	NS	PS	PD
1385	1127	1127	845	824	869

**Table 3 plants-11-03097-t003:** Proposed unsupervised techniques in our study.

Techniques (1)	Abbreviation (1)	Clusters (2)	Abbreviation (2)	Proposed Cluster (3)	Abbreviation (3)
K-Means	M1	Cluster 1	C1	Elbow based four-grayscale	EG
K-Means++	M2	Cluster 2	C2	Silhouette based six-grayscale	SG
Minibatch K-means	M3	Cluster 3	C3	Elbow based four-threshold	ET
Spectral	M4	Cluster 4	C4	Silhouette based four- threshold	ST
Birch	M5	Cluster 5	C5	Elbow based five-colored	EC
				Silhouette based eight- colored	SC

## Data Availability

All data generated or analyzed during this study are included in this published article.

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
