# Peer review of "A Multi-Considered Seed Coat Pattern Classification of *Allium* L. Using Unsupervised Machine Learning"

_plants, 2022, doi:10.3390/plants11223097_

Round 1
Reviewer 1 Report
I have reviewed the paper " A multi-considered seed coat pattern classification of Allium 2 using unsupervised machine learning". -Abstract is well written. -Need and scope of work in the introduction section is missing. -Objectives should be more clear. -In results sections figures and graphs are not much clear. -In Figure 7, some images fade. - Discussion is not well written. It needs some improvement. - Conclusion should be precise. After these minor changes the paper should be accepted for publication.Author Response
Please see the attachment.

Reviewer 2 Report
The manuscript is interesting and provides new mechanisms and techniques to analyze the seed coat pattern in plants, a good trait to support phylogenetic and taxonomic studies in a great number of genera of flowering plants. However, I have found a set of deficiencies in the manuscript.
In the Abstract, you mentioned the selection of images from more than 100 Allium species, but along the text, I have not found any reference about the source of these images, the images of types of anticlinal and periclinal walls observed in human results, or if the images were observed under light or scanning electron microscope. Besides, the use of "O" and "UO" morphologies is uncertain, because precedent authors use the Greek letter omega in capital to describe these types of morphologies.
It is important to illustrate the types of anticlinal and periclinal walls, based on precedent papers and base on your own results, to understand the use of machine-based identification, and to provide the conclusions.
It is unclear if the studied seed images belong to mature seeds. In the study of these structures, it is very important the maturation of seeds to recognise the morphology of testa cells and their walls.
In Material and methods, the procedence of images is missing. In the preprocessing module, you discuss the use of color images, but if you analyzed SEM images, these are always in grayscale.
In the Introduction, the distribution range of the genus Allium is concerned to the northern hemisphere, but one species is an endemism from Southern Africa.
In the reference 28, data about the publication are missing. The paper was published within a series of paers in B. Mitschang et al. (Hrsg.): BTW 2017 – Workshopband, Lecture Notes in Informatics (LNI), Gesellschaft für Informatik, Bonn 2.
In the reference 47, the journal is uncertain, not Digital Object Identifier. According to IEEE Xplore the cite is: K. P. Sinaga and M. -S. Yang, "Unsupervised K-Means Clustering Algorithm," in IEEE Access, vol. 8, pp. 80716-80727, 2020, doi: 10.1109/ACCESS.2020.2988796.
In the reference 48, the journal is Proceedings of the VLDB Endowment.
Reviewer 3 Report
The study is interesting, but it is necessary to explain clearly and rewrite some aspects related with the introduction, results, discussion, and conclusions.
Principally, the authors don’t include the source of the material used in their study, and the results and conclusions are unclear. I suggest including a comparative table with the differences observed among the human and machine studies.
In the attached file, I have noted several comments and suggestions.
It is necessary to revise the following points:
- Page 2 (line 74 to 97). The authors explain methodology and the followed steps. I suggest finishing the introduction with a clear explanation of the objectives of this study. It is necessary to know all the previous knowledge of the genus Allium in the introduction. On the other hand, I suggest including a comparative table with the diagnostic features proposed by previous authors.
- Page 3 (Table 1).
- Page 4. I suggest including a table with this information, which help to synthesize all the data.
- Page 5 (line 173-175)
- Page 8 related with the figure 7.
- Page 9, 10.
- Page 11. It is necessary to include the source of the images and if all of these are from SEM.
- Pages 12-14

Reviewer 4 Report
Congratulations on a great article!
Round 2
Reviewer 2 Report
I have revised the new version, with the suggested corrections. The supplementary files and the new figures provide the precedent manuscript, and contribute to understand the study.
I consider that this new version must be accepted to be published in the journal.
Reviewer 3 Report
Thank you for your revision, the text is clearer now.
All the suggestions, corrections and comments have been reviewed.
It needs only to check in the text (line 279), in the table 3 and in the figure 9 the use of UO or Ou, because it is written in both forms along the text.
